# Targeted innate immune inhibition therapy compared with antibiotics for recurrent acute cystitis: a randomized, open-label phase 2 trial

**A list of authors and their affiliations appears at the end of the paper**

Cystitis is a bacterial infection of the bladder that occurs in about half of women at least once in their lifetime. Antibiotics such as nitrofurantoin are used to treat cystitis, but antibiotic resistance is a concern, especially for recurrent infections. Here we report an open-label, randomized, single-centre, phase 2 study to analyse the acute and long-term safety and efficacy of the IL-1 receptor antagonist anakinra, compared with nitrofurantoin, in recurrent cystitis. A total of 30 adult female patients with a documented history of recurrent cystitis and a current acute cystitis episode were randomized in a 2:1 ratio to treatment with anakinra ($n = 20$) or nitrofurantoin ($n = 10$) for 5 days. Primary and secondary efficacy end-points were reached, defined as the reduction in typical symptoms, measured by the acute cystitis symptom score (day 5), longitudinal symptom scores, recurrence rates, quality of life, gene expression analysis and microbiology at follow-up on days 15 and 30 and at 6 months. Symptom scores were decreased in the anakinra ($P < 0.001$) and nitrofurantoin ($P < 0.001$) arms after 5 days and remained low after 15 days, 30 days and 6 months. Recurrences were less frequent after 6 months in both treatment groups compared with the 6-month pre-enrolment history ($P < 0.001$ for anakinra and $P = 0.004$ for nitrofurantoin), and the quality of life was increased, without adverse effects. Immune gene expression was rapidly inhibited in the anakinra-treated patients but not in the nitrofurantoin group. Targeted innate immune inhibition therapy shows non-inferiority to nitrofurantoin in patients with recurrent acute cystitis. German Clinical Trials Register ID: DRKS00025964.

Rather than directly killing the bacteria with antibiotics, it should be possible to rebalance the often excessive and damaging immune response in infected patients and restore their antibacterial defence[1]. New molecular insights suggest that this is the case, as experimental studies have detected potent treatment effects of innate immune inhibitors, against antibiotic-sensitive and antibiotic-resistant bacteria[2–6], offering an interesting new treatment approach and alternative to antibiotics.

Acute uncomplicated cystitis, a bacterial infection of the urinary bladder, affects about 50–60% of all women during their lifetime[7,8]. A recent survey in the USA and Germany showed that acute cystitis may become debilitating, as 16–43% of the patients did not have complete symptom relief after initial therapy[9] and 33–37% experienced recurrent infections. The treatment of urinary tract infection (UTI) has become increasingly problematic, owing to the rapid increase in antimicrobial resistance in bacteria causing UTI[10,11], and especially in patients with

✉e-mail: ines.ambite@med.lu.se

**Fig. 1 | CONSORT flow diagram.** Consolidated Standards of Reporting Trials (CONSORT) flow diagram of the randomized, open-label, phase 2 trial in patients with a history of recurrent cystitis and an acute episode. Patients were randomized to immunomodulatory (anakinra, $n = 20$) or antibiotic (nitrofurantoin, $n = 10$) treatment.

recurrent infections[12]. Finding alternative therapies and reducing antimicrobial use are therefore essential, but previous non-antibiotic strategies have resulted in poorer clinical outcomes, compared with antibiotic treatment[13].

Studies in experimental models have identified IL-1β over-activation as a cause of acute cystitis, leading to hyperinflammation of the urinary bladder and impaired bacterial clearance[2]. Treatment with the IL-1 receptor antagonist (IL-1RA) anakinra, which inhibits the effects of IL-1α and IL-1β, was shown to protect mice against acute cystitis and to accelerate bacterial clearance from the bladder, with a similar efficacy to antibiotics[2,5], identifying IL-1 inhibition as an approach with therapeutic potential. This study analysed the acute and long-term efficacy and safety of innate immune inhibition in adult female patients with an acute episode of uncomplicated cystitis and a history of recurrent UTI. The patients were randomized to treatment with the IL-1RA anakinra (Kineret) or the antibiotic nitrofurantoin. The results suggest that therapy blocking the IL-1 receptor may be a precise, efficient and safe way to treat acute cystitis and prevent recurrent infection, with similar efficacy to antibiotics.

## Results

This prospective randomized study was conducted between 2 September 2021 and 3 September 2024, to compare the therapeutic efficacy of the IL-1RA anakinra to that of nitrofurantoin in recurrent acute cystitis (Fig. 1). The study was an open-label, randomized, single-centre, two-arm, parallel-group, phase 2 trial in female patients with a history of recurrent cystitis and an ongoing acute episode of cystitis defined by typical symptoms graded using the established acute cystitis symptom score (ACSS) method (http://ACSS.world, questionnaire in Supplementary Fig. 1)[7,14,15]. A diagnosis of acute cystitis was defined by a sum score ≥6 for the typical symptoms at visit 1 (refs. 16,17), according to a predefined scale[7] for each symptom (frequency, urgency, burning urination pain, incomplete bladder emptying, lower pelvic pain, haematuria), supported by urine microbiology and neutrophil counts (Fig. 2a). Of the 33 patients invited to participate, 30 were enrolled, randomized to anakinra ($n = 20$) or nitrofurantoin ($n = 10$) treatment and completed the study (Fig. 1). Three approached patients were not enrolled in the study owing to the lack of informed consent ($n = 1$) or for

not meeting inclusion criteria ($n = 2$). The patients were randomized to receive either anakinra treatment, as five daily subcutaneous injections at the study clinic, or oral nitrofurantoin antibiotic treatment for 5 days (Table 1).

### Primary outcome variables

The response to treatment was first analysed based on the change in typical symptom scores in each patient (Fig. 2b) on day 5 post-treatment compared with enrolment. Treatment reduced the typical symptom scores in both groups on day 5, defined by pairwise analysis of each patient ($P < 0.001$; Fig. 2c). A typical symptom score below that at enrolment was identified in 19 out of 20 patients in the anakinra group and 8 out of 10 patients in the nitrofurantoin group (Fig. 2b). The range of typical symptom scores at enrolment did not differ between the anakinra group (7–15) and the nitrofurantoin group (6–15) ($P = 0.673$; Table 1 and Supplementary Fig. 2).

The response to treatment was subsequently evaluated based on symptom relief dynamics, defined by the ACSS method. The symptom relief dynamics were quantified into five categories as all, most or some symptoms gone; no change; or worsening of symptoms after treatment, recorded individually at each follow-up time point and defined relative to enrolment (Fig. 2d). In the anakinra treatment group, 60% of the patients reported a loss of most or all symptoms at the end of treatment. The corresponding response frequency was 70% in the nitrofurantoin treatment group. There was no significant difference in symptom dynamics between the anakinra and nitrofurantoin treatment groups ($P = 0.700$ on day 5). A significant improvement of the quality-of-life score was also recorded on day 5, due to a reduction in discomfort, interference with daily activities and social activities (Fig. 2e,f; $P < 0.001$ and $P = 0.005$).

### Secondary analysis of long-term follow-up of the treatment effect

A prolonged treatment effect was detected in both treatment groups, with some variation. A reduction in typical symptom scores was recorded on days 15 and 30 and after 6 months (Fig. 2c; all $P < 0.001$ compared with enrolment). A lasting effect on symptom relief dynamics was also reported on days 15 and 30 and after 6 months (Fig. 2d;

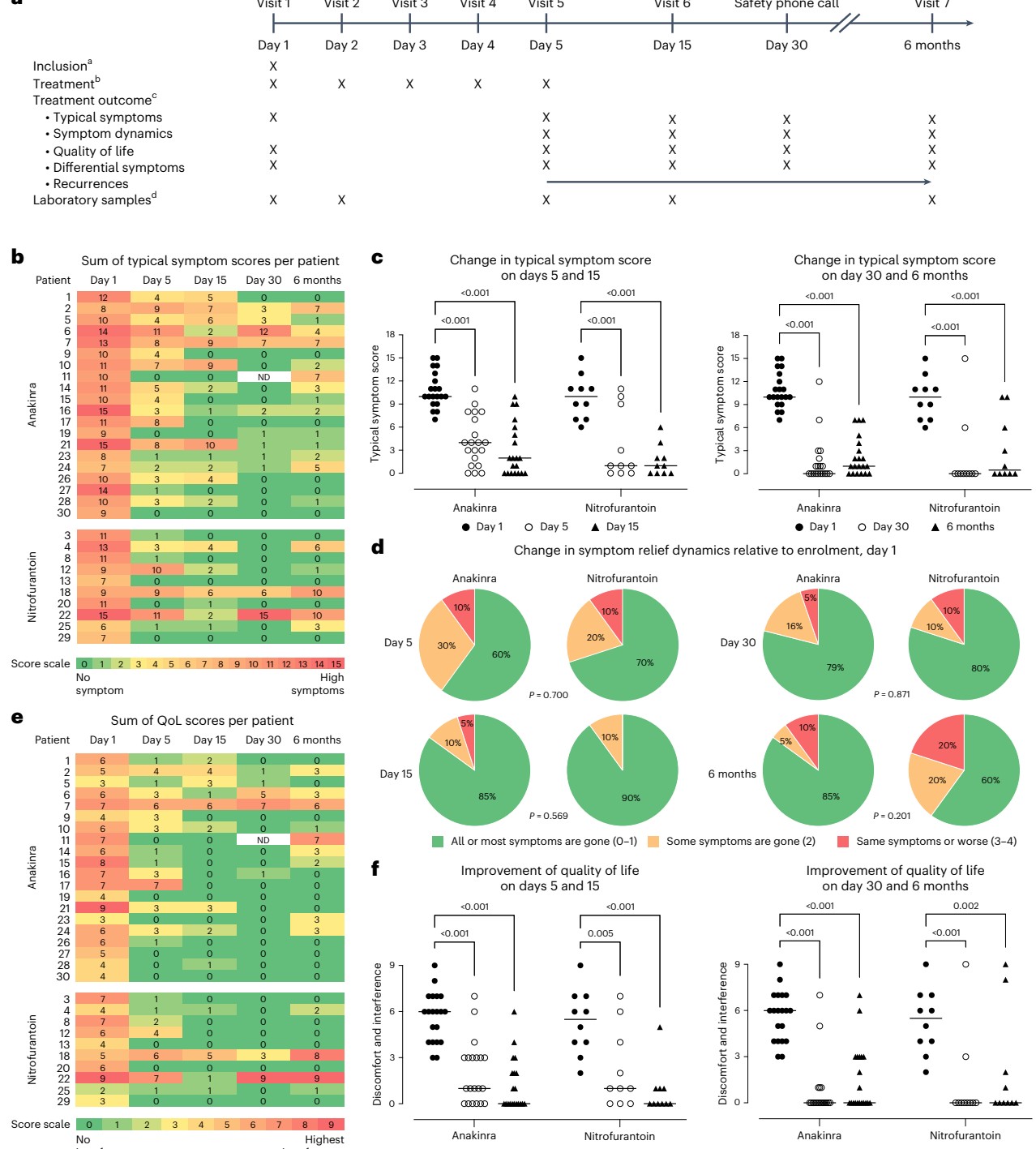

**Fig. 2 | Study protocol and treatment efficacy−individual treatment responses defined by typical symptom scores, symptom dynamics and quality of life. a**, Schedule of assessment. [a]Inclusion criteria: history of recurrent UTI and ongoing acute episode. Typical symptom (frequency, urgency, burning urination pain, incomplete bladder emptying, lower abdomen pain, haematuria) score ≥6. Supported by urine microbiology and neutrophil counts. [b]Treatment: anakinra (Kineret, 100 mg daily), 20 patients, or nitrofurantoin (Uro-Tablinen, 100 mg × 2 daily), 10 patients. [c]Treatment outcomes: typical symptoms (frequency, urgency, burning urination pain, incomplete bladder emptying, lower abdomen pain, haematuria), quality of life (rating of discomfort, disruption of daily activities, disruption of social activities), differential symptoms (flank pain, vaginal discharge, urethral discharge, fever), symptom dynamics (change in symptoms compared with enrolment), recurrences defined using the same criteria as for inclusion. [d]Laboratory samples: urine analysis, blood RNA samples and

differential counts. **b**, Heat map of typical symptom scores in individual patients, arranged by treatment group. **c**, Comparative analysis of typical symptom scores at enrolment and at different times of follow-up. **d**, Treatment efficacy evaluated as symptom relief dynamics. Improvement was detected at all time points. Two-sided P values from the Cochran−Armitage trend test comparing treatment groups are reported. **e**, Heat map of quality-of-life scores in individual patients, arranged by treatment group. **f**, Comparative analysis of quality-of-life scores at enrolment and at different times of follow-up. For **c** and **e**, two-sided adjusted P values from estimated marginal means post hoc comparisons are reported (see Supplementary Fig. 2 for ordinal logistic mixed-effects model test results). Horizontal lines indicate group median values. ND, no data; in the anakinra group, n = 20 for days 1, 5 and 15 and 6 months, and n = 19 for day 30; in the nitrofurantoin group, n = 10 per time point.

## Table 1 | Demographic data

| Patient characteristics | Anakinra (n=20) | Nitrofurantoin (n=10) |
|---|---|---|
| **Age** | | |
| Median (range) (years) | 26 (22–71) | 25 (19–52) |
| <20, n (%) | 0 (0) | 1 (10) |
| 20 to <35, n (%) | 16 (80) | 7 (70) |
| ≥35, n (%) | 4 (20) | 2 (20) |
| **WOCBP** | | |
| Yes, n (%) | 18 (90) | 9 (90) |
| No, n (%) | 2 (10) | 1 (10) |
| **BMI** | | |
| Median (range) (kg m$^{-2}$) | 21.9 (18.5–60.4) | 22.4 (19.5–34.4) |
| <30, n (%) | 19 (95) | 9 (90) |
| ≥30, n (%) | 1 (5) | 1 (10) |
| **ACSS typical domain score (visit 1)** | | |
| Median (range) | 10 (7–15) | 10 (6–15) |
| **History of recurrent cystitis, number of episodes** | | |
| Past 6 months, median (range) | 2.5 (1–10) | 4 (2–8) |
| Past 12 months, median (range) | 6 (3–20) | 7.5 (3–16) |

WOCBP, woman of childbearing potential; BMI, body mass index, weight (kg) divided by the square of height (m).

nonsignificant for every time point). A reduction in typical symptom scores (Fig. 2b) below the enrolment cut-off was observed in 15 out of 20 patients in the anakinra group on day 15 and 18 out of 20 on day 30, and in 9 out of 10 patients in the nitrofurantoin group on day 15 and 8 out of 10 on day 30. This positive effect of treatment on the typical symptom score remained significant after 6 months in both groups ($P < 0.001$).

The quality of life was significantly improved long term, reflecting the reduction in symptoms, discomfort and interference in individual patients (Fig. 2f) ($P < 0.001$ on days 5, 15 and 30 in the anakinra group compared with enrolment). A similar trend was observed in the nitrofurantoin treatment group ($P = 0.005$, <0.001 and <0.001 respectively). This positive effect of treatment on the quality of life remained significant after 6 months in both groups ($P < 0.001$ in the anakinra group and $P = 0.002$ in the nitrofurantoin group).

### Secondary analysis of treatment effect on differential symptoms and recurrences

In addition to the primary outcome variables, the frequency of differential symptoms was evaluated (Fig. 3a). These included flank pain, fever, and vaginal and/or urethral discharge, potentially indicating more severe disease or an infection other than cystitis. A significant reduction in differential symptom scores was observed in both treatment groups ($P = 0.043$ on day 5 and $P = 0.023$ on day 15 in the anakinra group compared with enrolment, and $P = 0.032$ on day 5 and $P = 0.005$ on day 15 in the nitrofurantoin group compared with enrolment; Supplementary Fig. 2).

Furthermore, a significant reduction in recurrence rates was observed in both treatment groups during the 6 months follow-up after inclusion (Fig. 3b). The number of episodes documented before enrolment and after treatment was compared for each patient. The recurrence range was 0–2 in the anakinra group ($P < 0.001$) and 0–3 in the nitrofurantoin group ($P = 0.004$). The number of documented episodes at enrolment and after treatment did not differ between the anakinra and nitrofurantoin treatment groups ($P = 0.081$ and $P = 0.484$, respectively).

### Exploratory analysis of systemic disease response to treatment

The disease response and treatment effects were further investigated in peripheral blood RNA samples, obtained at enrolment and after 2 days or 5 days of treatment. Gene expression analysis revealed a remarkable inhibitory effect in the anakinra treatment group, after 2 days and 5 days but not in the nitrofurantoin-treated patients (Fig. 3c). The IL-1 response was inhibited by anakinra treatment, as well as IL-6, TNF and IFNγ. The cytokine storm signalling pathway was strongly inhibited on day 5 after anakinra treatment, including the IL-1-dependent response, as well as genes involved in neutrophil degranulation and toll-like receptor signalling. By contrast, there was no effect on these pathways in the nitrofurantoin treatment group (Fig. 3d), identifying immune inhibition as a specific effect of anakinra treatment.

The systemic response to treatment was subsequently investigated by analysing the leukocyte count, as well as the differential neutrophil and lymphocyte counts (Extended Data Fig. 1). Anakinra treatment reduced leukocyte and neutrophil counts at days 2 and 5 compared with enrolment ($P < 0.001$ and 0.001, respectively). A reduction was also observed in the nitrofurantoin group ($P = 0.008$ and 0.004, respectively). The effects had subsided at follow-up on day 15. Lymphocyte counts were not significantly affected by treatment.

### Secondary analysis of microbiological findings

Quantitative urine cultures identified a complex flora, and ≥10$^3$ colony forming units (CFU) ml$^{-1}$ was selected as a cut-off for positive cultures (Extended Data Table 1)[18]. *Escherichia coli* was the predominant organism at enrolment (11 out of 20 patients in the anakinra arm and 5 out of 10 patients in the nitrofurantoin arm). Eight of these patients also carried other bacterial species. An additional six patients in the anakinra arm and four in the nitrofurantoin arm were defined as carrying a mixed flora with cultures positive for *Klebsiella*, enterococci or streptococci (Extended Data Tables 2 and 3). Three patients in the anakinra arm and one patient in the nitrofurantoin arm had <10$^3$ CFU ml$^{-1}$ in the standard culture at visit 1.

Anakinra treatment was associated with a long-term effect on the bacterial burden, with 9 out of 20 (45%) patients showing <10$^3$ CFU ml$^{-1}$ after 6 months. The corresponding frequency was 3 out of 10 (30%) in the nitrofurantoin group. On day 5, 3 out of 19 (16%) patients in the anakinra group and 7 out of 10 (70%) in the nitrofurantoin group were culture negative (<10$^3$ CFU ml$^{-1}$), and on day 15, 6 out of 20 (30%) patients in the anakinra group and 3 out of 9 (33%) patients in the nitrofurantoin group showed negative urine cultures (<10$^3$ CFU ml$^{-1}$).

A subgroup analysis was performed in patients with bacterial numbers in urine ≥10$^5$ CFU ml$^{-1}$ at enrolment (12 patients in the anakinra arm and 8 patients in the nitrofurantoin arm). The treatment effect was confirmed in these patients, comparing the typical symptom score and quality of life, before the onset of treatment on day 1 to the last day of treatment, day 5 (Fig. 4; $P < 0.001$ for both groups). The treatment effect was also significant in the patients, who had bacterial counts <10$^5$ CFU ml$^{-1}$ ($P = 0.008$).

### Safety analysis

Anakinra has been extensively used and has a well characterized safety profile[19]. Common side effects of nitrofurantoin include nausea, vomiting, diarrhoea, loss of appetite, headaches and dizziness or feeling sleepy. The safety analysis of patients receiving at least one dose of either study drug is summarized in Table 2. There were no severe adverse events (AEs) in either arm of the study. One AE was defined as possibly related to study drug administration in the anakinra group (for example, vaginal rash in a genital herpes simplex virus-positive patient after 5 days). In total, 7 out of 20 patients (35%) in the anakinra group and 1 out of 10 patients (10%) in the nitrofurantoin group had at least one AE. All AEs were mild in severity in both treatment groups, and no AE led to study discontinuation (Table 2 and Extended Data Table 4).

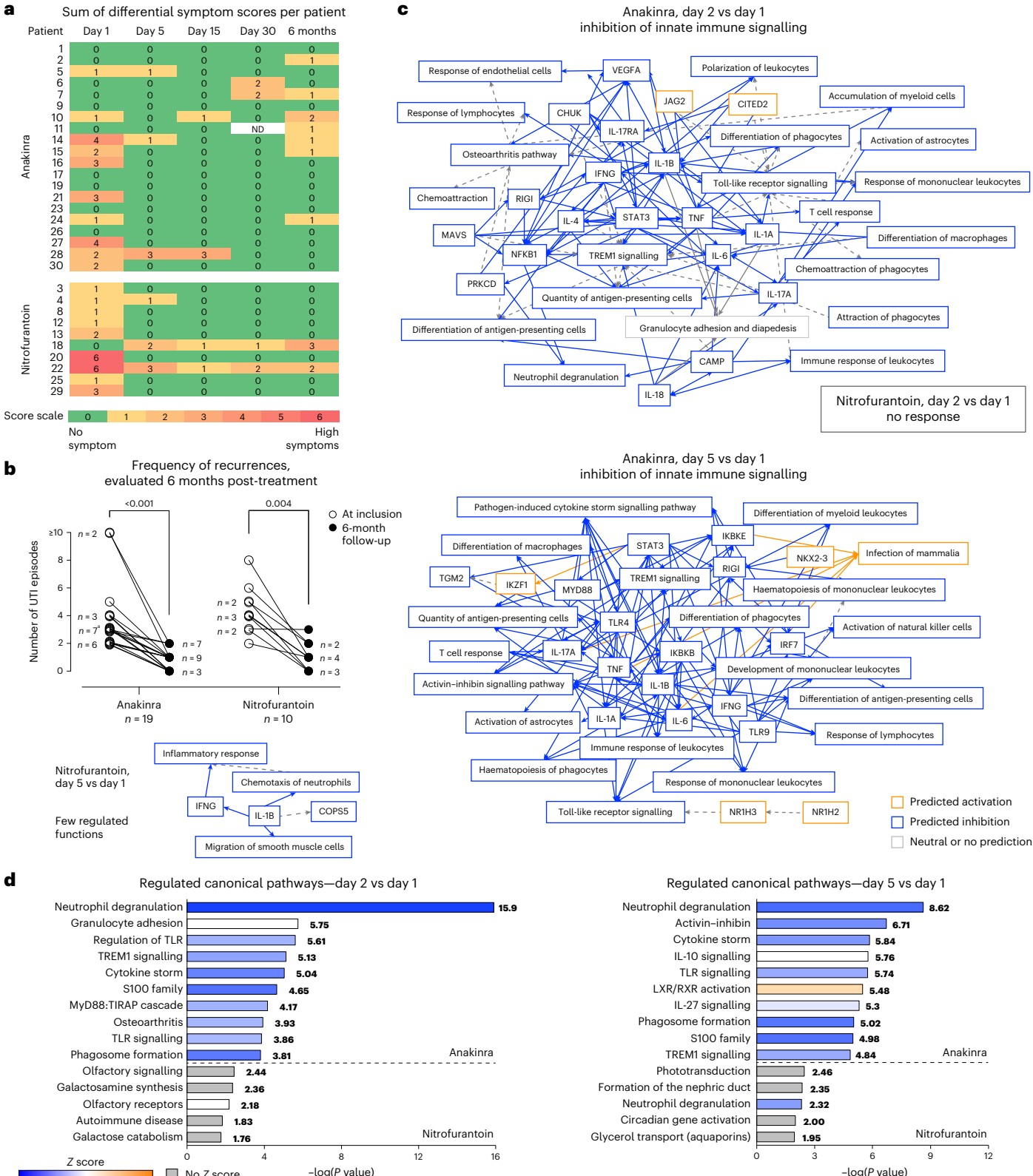

**Fig. 3 | Treatment efficacy—differential symptoms, recurrences and gene expression analysis. a**, Heat map of differential symptom scores in individual patients, arranged by treatment group. A complete statistical analysis is shown in Supplementary Fig. 2. **b**, Frequency of cystitis recurrences after treatment with anakinra or nitrofurantoin in individual patients compared with their previous 6-month history of cystitis episodes. The number of patients for each data point is indicated. ªThree patients had a medical history of three documented episodes in the past year. Two-sided *P* values from Wilcoxon matched-pair signed rank test

are reported. **c**, Genome-wide transcriptomic analysis from peripheral blood RNA. Functional networks of the transcriptomics response in the anakinra group, on day 2 (*n* = 17) and day 5 (*n* = 19) compared with day 1 (*n* = 19). No functional network was predicted in the nitrofurantoin group on day 2 (*n* = 9) compared with day 1 (*n* = 10), and limited response on day 5 (*n* = 9). **d**, Top regulated pathways on day 2 and day 5 in anakinra- and nitrofurantoin-treated patients. Right-tailed Fisher exact test −log(*P* values) are indicated.

## Discussion

The main goal of antibiotic therapy is to remove bacteria and end the disease processes that they provoke. Targeting the innate immune response and enhancing its antibacterial effects offer an alternative therapeutic option[3,5]. This approach requires insights into the cause of disease and the key molecules that should be targeted to inhibit destructive inflammatory cascades. In this study, the efficacy of the IL-1RA anakinra was compared with that of the antibiotic nitrofurantoin, showing similar efficacy of both treatment approaches in patients with recurrent cystitis.

IL-1β over-activation has been identified as a molecular basis of severe cystitis, in experimental models of acute cystitis, and associated with genetic dysfunctions enhancing IL-1β processing[2,4,5]. Mice lacking IL-1β (*Il1b*[−/−]) are protected against acute cystitis[2], suggesting that disease is driven by IL-1β and may be avoided or attenuated if activation is inhibited and has been achieved using IL-1RA treatment, which inhibits the effects of IL-1α and IL-1β. This study detected a substantial effect of IL-1RA on symptom scores, urine microbiology, recurrences and quality of life, supporting a role of IL-1 activation also in recurrent cystitis. Gene expression analysis detected an extensive effect of anakinra treatment on innate immunity, with inhibition of the cytokine storm response as well as key regulators of innate immunity. While long-term anakinra treatment has been associated with an increased rate of serious infections[20], anakinra treatment was limited to 5 days and there was no evidence of lasting innate immune inhibition at the 6-month follow-up. The short anakinra treatment used in this study was therefore not expected to impair the antibacterial defence of treated patients.

Patients with acute cystitis develop a range of typical symptoms, which are used to diagnose individual episodes. The molecular basis of recurrent cystitis remains less clear, as recurrent infections are expected to be accompanied by complex immune responses, probably reflecting repeated immune activation by infection and resulting chronicity[16,17,21]. The ACSS is a self-reporting symptom questionnaire used to assess the severity of symptoms in women with symptomatic lower urinary tract infections and their impact on quality of life, with the possibility to monitor treatment efficacy. The ACSS method is well accepted and has been shared between investigators internationally to standardize study design and achieve reproducibility in this complex field[16,17,21]. In the ACSS validation studies, the best balance between sensitivity and specificity, with the highest Youden's index and area under the curve (AUC), was found when using a minimum summary score of 6 for the 'typical domain', resulting in a sensitivity of 0.87 and a specificity of 0.88 for the presence of acute cystitis[22]. Many of these symptoms emanate from the local innate immune response to infection, including pain, which is triggered by direct infection of mucosal nerve cells as well as IL-1 from adjacent epithelial cells, supporting the suitability of anakinra for this patient group[4].

The urine cultures showed a level of complexity that would be expected in patients with recurrent acute cystitis[23]. A study has

shown that low-level bacteriuria is frequent in cystitis, with up to 15% of patients having *E. coli* < 10³ CFU ml⁻¹ in voided midstream urine[18]. This compares with our study in which 4 patients had <10³ CFU ml⁻¹ in voided midstream urine at visit 1. Significant bacteriuria defined by ≥10⁵ CFU ml⁻¹ was detected in 12 patients in the anakinra group and 8 patients in the nitrofurantoin group. A subgroup analysis of those patients confirmed to the overall treatment effect on day 5, with a reduction in typical symptom scores, improved quality of life and reduced recurrence rates. Furthermore, lasting effects on the flora were suggested by reductions in bacterial numbers after 6 months.

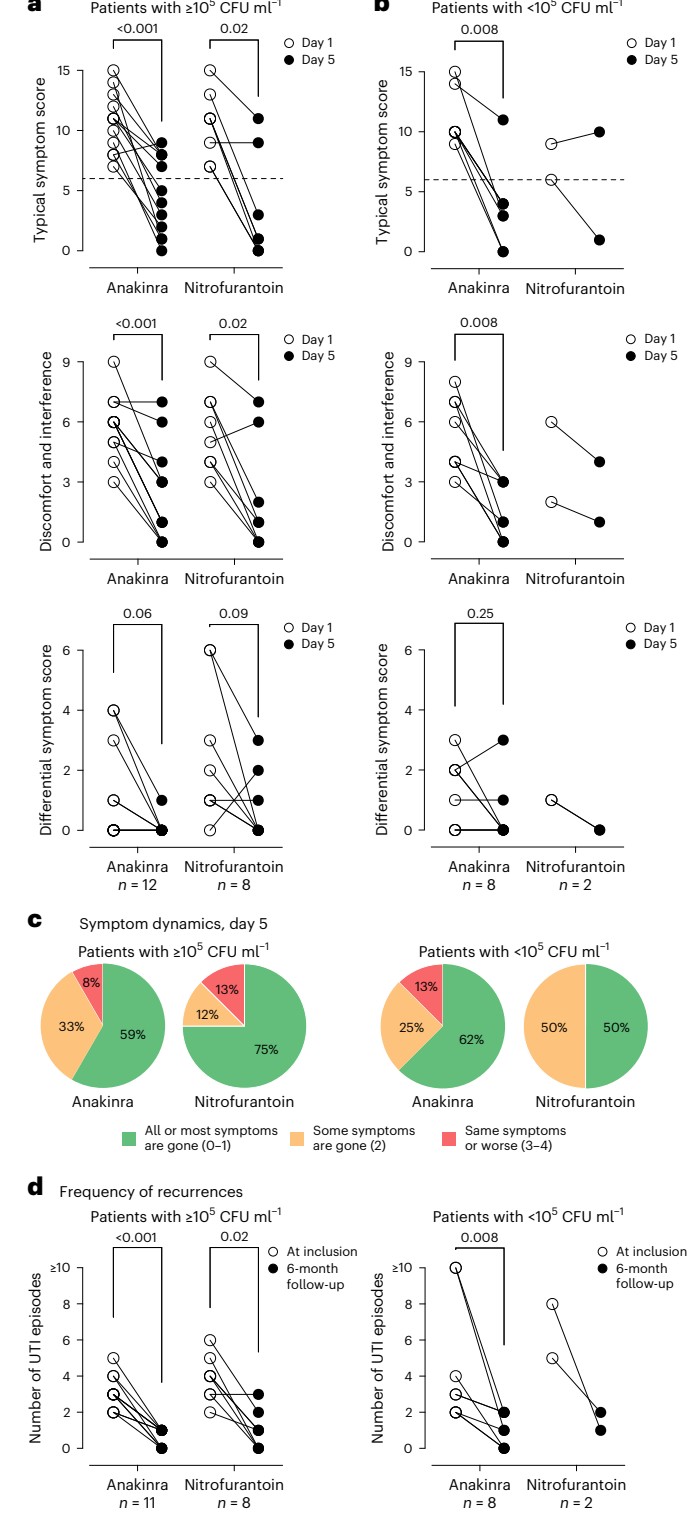

**Fig. 4 | Sub-analysis of patients with ≥10⁵ CFU ml⁻¹ of urine. a,** Comparative analysis of the typical symptom, quality-of-life and differential symptom scores at enrolment (day 1) and after treatment (day 5) in patients with ≥10⁵ CFU ml⁻¹ in urine cultures. *n* = 12 for anakinra; *n* = 8 for nitrofurantoin. **b,** Comparative analysis of the typical symptom, quality-of-life and differential symptom scores at enrolment (day 1) and after treatment (day 5) in patients with <10⁵ CFU ml⁻¹ in urine cultures. *n* = 8 for anakinra; *n* = 2 for nitrofurantoin. **c,** Treatment efficacy evaluated as symptom relief dynamics reported on day 5 in patients with ≥10⁵ CFU ml⁻¹ or <10⁵ CFU ml⁻¹ in urine cultures. **d,** Frequency of cystitis recurrences after treatment with anakinra or nitrofurantoin, in patients with ≥10⁵ CFU ml⁻¹ or <10⁵ CFU ml⁻¹ in urine cultures, compared with the history of infections during 6 months before enrolment in this study. *n* = 11 and 8 for anakinra; *n* = 8 and 2 for nitrofurantoin. For **a**, **b** and **d**, two-sided *P* values from Wilcoxon matched-pairs signed rank test are reported, except for nitrofurantoin patients with <10⁵ CFU ml⁻¹ in urine cultures.

**Table 2 | Adverse event summary**

| Treatment | Patients, n (%) | | | AEs[a], n | |
|---|---|---|---|---|---|
| | SAE[b] | AEs, mild | No AE | Possibly related | Not related |
| Anakinra | 0 (0) | 7 (35) | 13 (65) | 1 | 8 |
| Nitrofurantoin | 0 (0) | 1 (10) | 9 (90) | 0 | 1 |

[a]Individual events are listed in Extended Data Table 4. [b]SAE, serious adverse event.

The anakinra and nitrofurantoin treatments both reduced bacterial numbers, as well as recurrence rates, over time.

Anakinra showed non-inferiority to nitrofurantoin for end-points defined by the typical symptom score, quality of life and number of recurrences. While the study needs to be repeated and extended, consequences of implementing innate immune inhibition therapy may be discussed. Acute cystitis is one of the most common bacterial infections, affecting at least 50% of all women in their lifetime[8]. Replacing or reducing the need for antibiotics in these large populations would greatly reduce antibiotic use and side effects of antibiotics in individual patients. Furthermore, gene expression analysis suggested that these chronically infected patients have a systemic component to their immune response, which may maintain pain sensing and diffuse inflammation and contribute to the lingering symptoms between recurrent acute infection episodes. The effect of anakinra treatment on the immune response, which was not observed in the nitrofurantoin group, suggests that immunotherapy may provide additional benefits in this patient group and possibly in patients with infections caused by antibiotic-resistant bacteria.

## Methods

### Study design

For this randomized, open-label, single-centre, phase 2 trial, patients were enrolled at the Clinic for Urology, Pediatric Urology and Andrology in Giessen, Germany. The trial was registered on 27 July 2021 at the German Clinical Trials Register number DRKS00025964 (https://drks.de/search/en/trial/DRKS00025964) and EudraCT number 2019-004209-28. Diagnosis and treatment followed an established clinical pathway, and mandated routine tests included urinalysis, full blood count, and urine and blood sampling for molecular analyses (Fig. 2a). No formal sample size calculation evaluating the power of the trial has been performed. However, consideration regarding the sample size was made based on previous studies of anakinra in a murine acute cystitis model[2,5]. A total of 33 women were invited to participate in the study, but 3 were not enrolled as they did not fulfil the inclusion criteria, resulting in a study cohort of 30 participants; 20 participants were treated with anakinra and 10 participants with nitrofurantoin (Fig. 1).

The trial was conducted in accordance with the principles of the Declaration of Helsinki, the International Council for Harmonisation guidelines for good clinical practice, and applicable national laws and regulatory requirements. Ethical approval was obtained from the German Ethical Review Authority (ethics vote AZ10/21). Written information was presented to the patients, and participation required a signed informed consent.

### Patients

Female adult patients with an acute episode of recurrent uncomplicated cystitis were included in this study (Table 1). Enrolment was based on a sum score of the main UTI symptoms at visit 1 ≥ 6 as reported on the ACSS typical domain[7,14,15,24,25]. The ACSS is a self-reporting questionnaire for the clinical diagnosis of acute uncomplicated cystitis[7,14] and is also a proven instrument for patient-reported outcome assessment[25,26]. The ACSS method (see questionnaire in Supplementary Fig. 1, http://ACSS.world) has been developed as a diagnostic tool for uncomplicated UTI, to more precisely define patients with this disease state based on

reported outcome measurement[26]. It has been tested in trials comparing antibiotic versus non-antibiotic therapy, in which effects on clinical criteria are important as main objectives[15].

The typical symptom score cut-off ≥6 was used as a symptomatic inclusion parameter. Typical symptoms include frequency, urgency, burning pain at urination, incomplete bladder emptying, lower pelvic pain and haematuria; each symptom is evaluated according to a pre-defined Likert scale[7] (intensity score 0–3 and a maximum total score of 18). Typical symptoms must be acute, developed within ≤6 days of enrolment, and the patients must have a history of recurrent cystitis (≥3 documented episodes in the past year or ≥2 documented episodes in the past 6 months). Follow-up visits were performed on days 2, 5, 15 and 30 and after 6 months.

Inclusion and exclusion criteria stated in the protocol are as follows:

Inclusion criteria

- Stable patients with an acute episode of recurrent uncomplicated cystitis
- Sum score of the typical UTI symptoms at visit 1 ≥6, reported on the ACSS; urgency; frequency; burning at voiding; feeling of incomplete emptying; lower abdomen pain; blood in urine
- Typical symptoms acute, developed within ≤6 days
- History of recurrent cystitis (≥3 documented episodes in the history in the past year or ≥2 documented episodes in the history in the past 6 months)
- Female participants, 18–65 years old
- Signed written informed consent form
- Capability and willingness to comply with study procedures
- Negative urine pregnancy test in women of childbearing potential
- Contraception

Exclusion criteria

- Breastfeeding
- Signs of pyelonephritis
- Uncontrolled diabetes mellitus
- Neutropenia (<1.5 × 10^9 l^-1)
- Signs of genital infections (vaginitis, cervicitis)
- Anamnestic exclusion of patients with active severe infections, history of viral hepatitis
- Patients using Cytochrome P450 medications (for example, warfarin, phenytoin)
- Patients using phenytoin medication
- Extragenital conditions, nephrological conditions, urological conditions, urinary catheters that may lead to complicated UTI
- Severe uncontrolled systemic disease
- Systemic antibiotic therapy within 5 days before inclusion
- Impaired renal function (creatinine clearance <60 ml min^-1)
- Known allergies or contraindications to Kineret and nitrofurantoin Uro-Tablinen
- Malign diseases
- Immunosuppression
- Previously enrolled in this trial
- Pathological liver enzymes
- Polyneuropathies
- Glucose-6-phosphate dehydrogenase deficiency
- Participation in other interventional clinical trials
- Use of diclofenac and ibuprofen analgetic medication while participating in this clinical trial

### Trial procedures

All enrolled participants were randomized to two study treatment groups at visit 1; the chance for allocation to the anakinra group or nitrofurantoin group was 2:1. Randomization was performed

by the central office of the Center for Clinical Trials of the Philipps-University Marburg. The administrated investigational product was then documented together with the date, time and study code in the source documentation and electronic Case Report Form (eCRF). The analysis of the study parameters was blinded. Patients randomized to anakinra were allocated five prefilled syringes with 100 mg per 0.67 ml solution for injection (Kineret, Sobi), procured from the University Hospital Giessen and Marburg Clinic Pharmacy. Anakinra was administered by subcutaneous injection of 100 mg once a day for 5 days, given at around the same time each day. Patients receiving nitrofurantoin (Uro-Tablinen, Zentiva), procured from the University Hospital Giessen and Marburg Pharmacy Marburg, or with a prescription after 6 July 2023 owing to supply shortage, were instructed to take the drug orally at home on their own twice a day (2 × 100 mg) for 5 days. Proper documentation, including accompanying documents, was used to ensure that the antibiotic nitrofurantoin was used properly.

## Outcomes

The overall objective of the trial was to evaluate the efficacy and safety of anakinra for the treatment of acute cystitis, compared with conventional antibiotic treatment with nitrofurantoin, using a randomized protocol.

The primary efficacy end-point was the change in symptom score after 5 days of treatment, measured by the ACSS method.

Safety end-points included change in laboratory safety variables, vital signs, physical examination from baseline and the incidence of adverse events, treatment-emergent adverse events and serious adverse events during the trial.

Secondary end-points were the presence of bacteriuria and leukocyturia at each visit up to week 26 of the study and UTI recurrences up to 6 months.

Exploratory end-points included urine proteomics, gene expression analysis and DNA sequence analysis.

## ACSS

The ACSS (see questionnaire in Supplementary Fig. 1, http://ACSS. world) is a patient self-reporting questionnaire consisting of two parts: diagnostic part A and follow-up part B. Part A contains 18 items, allocated to 4 domains: 6 items on typical symptoms of acute cystitis ('typical' domain), 4 items for differential diagnosis ('differential' domain), 3 items on quality of life ('QoL' domain) and 5 items on additional conditions that may affect therapy ('additional' domain). Each item of the first three domains (typical, differential and QoL) is fitted with a 4-point Likert-type scale for assessing the severity of each symptom ranging from 0 (no symptom or discomfort) to 3 (severe symptom or discomfort). The 'additional' domain contains dichotomous 'yes or no' questions. Furthermore, part B includes an additional 'dynamics' domain formed by a question about the general evolution and changes in symptoms[22].

## Urine cultures and tests

Urine samples were analysed by quantitative bacterial cultures and dip slide for haematuria and leukocytes.

Midstream urine samples were cultured on cystine–lactose–electrolyte-deficient agar, Columbia nalidixic acid agar, MacConkey agar and Sabouraud agar (Thermo Fisher). Negative urine was <$10^3$ CFU ml$^{-1}$. Antibacterial resistance of the isolates was identified by automated antimicrobial susceptibility testing (VITEK 2, Biomerieux), followed by disk diffusion test using the European Committee on Antimicrobial Susceptibility Testing (EUCAST) break-point criteria.

## Blood samples

Analysis of white blood cell counts, including differential counts, was performed on venipunctured blood samples.

## Gene expression analysis

RNA, stabilized and purified from peripheral blood using Tempus blood RNA tubes and a purification kit (Applied Biosystems), was subjected to expression microarray analysis: 10 ng of total RNA was amplified and fragmented using the GeneChip 3' IVT Pico labelling kit and hybridized onto Human HT Clariom S arrays (all Thermo Fisher Scientific). Samples were processed in the GeneTitan System following the manufacturer's instruction (Thermo Fisher Scientific). Transcriptomic data were normalized using the robust multi-chip analysis algorithm implemented in the Transcriptome Analysis Console software (TAC v.4.0.1.36, Applied Biosystems, Thermo Fisher Scientific). A probeset (gene) is considered expressed if ≥50% of the samples in the dataset have a gene level detected above background value < 0.05. Fold change was calculated by comparing each treatment group with RNA obtained at the time of diagnosis (day 1). Differentially expressed genes and regulated pathways were analysed using Ingenuity Pathway Analysis software (IPA v.01-23-01, Qiagen).

## Statistical analysis

The typical symptoms, quality of life and differential symptoms were analysed using the ordinal logistic mixed-effects model (or cumulative link mixed model) fitted with the Laplace approximation. Typical symptom, quality-of-life and differential symptom score values were used as unstructured thresholds to calculate the cumulative odds ratio, with the assumption of proportional odds across thresholds, and every time point was included as a repeated ordinal outcome. This was followed by estimated marginal mean (or least-squares means) post hoc comparisons with Tukey's P value adjustment for comparing a family of five estimates, with ordinal dependent variables, repeated measures over time, between-subject factor (treatment) and random effect (subject). Treatment effects between time points within each arm, as well as differences between treatments at each time point, were evaluated (see statistical analysis results in Supplementary Fig. 2).

The symptom dynamics were analysed using the Cochran–Armitage trend test. Values are single ordinal measures of patient-reported outcome, independent of the other scores. Higher numbers mean worsening of symptom outcome compared with enrolment. The anakinra and nitrofurantoin groups were compared for each time point.

Recurrence rates were analysed by Wilcoxon matched-pairs signed rank test for each treatment group and Mann–Whitney test for each time point. Wilcoxon matched-pairs signed rank test was also used for the sub-analysis of patients with ≥$10^5$ CFU ml$^{-1}$ and patients with <$10^5$ CFU ml$^{-1}$, comparing day 1 and day 5.

Relative gene expression was analysed by ANOVA using the empirical Bayes method, and genes with a P value < 0.05 and an absolute fold change >1.5 were considered differentially expressed. P values for enrichment of canonical pathways and diseases and functions were calculated using a right-tailed Fisher's exact test with a null hypothesis of no non-random association. Activation Z scores were two sided.

For white blood cell counts and differential counts, normal distribution was tested using the D'agostino and Pearson, Anderson–Darling, Shapiro–Wilk and Kolmogorov–Smirnov tests. No complete dataset successfully passed a normality test. Data were therefore analysed using aligned rank transform ANOVA (see statistical analysis results in Supplementary Fig. 2). The aligned rank transform ANOVA model estimated significant changes across time points, which were compared using Wilcoxon matched-pairs signed rank test with the two-stage step-up method of Benjamini, Krieger and Yekutieli FDR (Q) = 1%, with non-parametric data, within-subject fixed effect (time), between-subject fixed effect (treatment) and random-effect intercept (subject).

Statistical analysis was performed using GraphPad Prism version 10.5.0 for macOS or R Studio version 2025.05.01 Build 513 (Posit Software, PBC) and R version 4.5.1 using the code available at Code Ocean (https://doi.org/10.24433/CO.8500789.v1)[27].

## Reporting summary

Further information on research design is available in the Nature Portfolio Reporting Summary linked to this article.

## Data availability

Gene expression data generated in this study are available at the National Center for Biotechnology Information Gene Expression Omnibus repository under accession GSE315861. All clinical data supporting the findings of this study and the study protocol are available in the Article and Supplementary Information. De-identified individual and/or study-level data will be shared with researchers who provide a methodologically sound proposal and if regulatory criteria are met. Access to anonymized data may be granted following review (time frame <20 office days) to ensure compliance with relevant ethical and legal considerations. Source data are provided with this paper. All other data that support the findings of this study are available from the corresponding author upon reasonable request.

## Code availability

Code used for statistical analysis is available at Code Ocean (https://doi.org/10.24433/CO.8500789.v1)[27].

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

## Acknowledgements

This study was funded by grants from the Swedish Research Council, Swedish Cancer Society, Österlund Foundation and the Royal Physiographic Society in Lund. Support for the Svanborg group was further provided by the European Union's Horizon 2020 research and innovation programme under grant agreement number 954360. The study was further supported by SelectImmune Pharma/Hamlet BioPharma.

## Author contributions

All authors contributed to data acquisition and writing (review and editing) and accessed the underlying study data. Study conceptualization and writing of the protocol were conducted by C.S., F.W., I.A. and G.G. Participant inclusion, investigation and data collection were conducted by F.W., A.P. and M.B.-H. Formal analysis was conducted by I.A., C.S., F.W. and S.A. All authors approved the final version of the paper and were responsible for the decision to submit it for publication.

## Funding

## Competing interests

I.A., S.A., G.G. and C.S. are part-time employees of Hamlet BioPharma and Lund University. I.A. and C.S. are shareholders in Hamlet BioPharma. Patents protecting the use of the IL-1RA for the

treatment of cystitis have been granted (EP3242680, with C.S. and I.A. as inventors). F.W. is a speaker of the DFG (German Research Foundation)-funded research group BARICADE (FOR5427/1-466687329) and a member of the DZIF (German Center for Infection Research; site: Giessen-Marburg-Langen). The other authors declare no competing interests.

## Additional information

**Extended data** is available for this paper at https://doi.org/10.1038/s41564-026-02262-1.

**Correspondence and requests for materials** should be addressed to Ines Ambite.

Ines Ambite ⓘ [1]✉, Adrian Pilatz ⓘ [2], Mareike Buch-Heberling[3], Shahram Ahmadi[1], Gabriela Godaly ⓘ [1], Florian Wagenlehner[2,4] & Catharina Svanborg ⓘ [1,4]

[1]Division of Microbiology, Immunology and Glycobiology, Department of Laboratory Medicine, Lund University, Lund, Sweden. [2]Clinic for Urology, Pediatric Urology and Andrology, Justus Liebig University Giessen, Giessen, Germany. [3]Coordinating Center for Clinical Trials, Department Giessen, Philipps-University of Marburg, Giessen, Germany. [4]These authors jointly supervised this work: Florian Wagenlehner, Catharina Svanborg. ✉e-mail: ines.ambite@med.lu.se

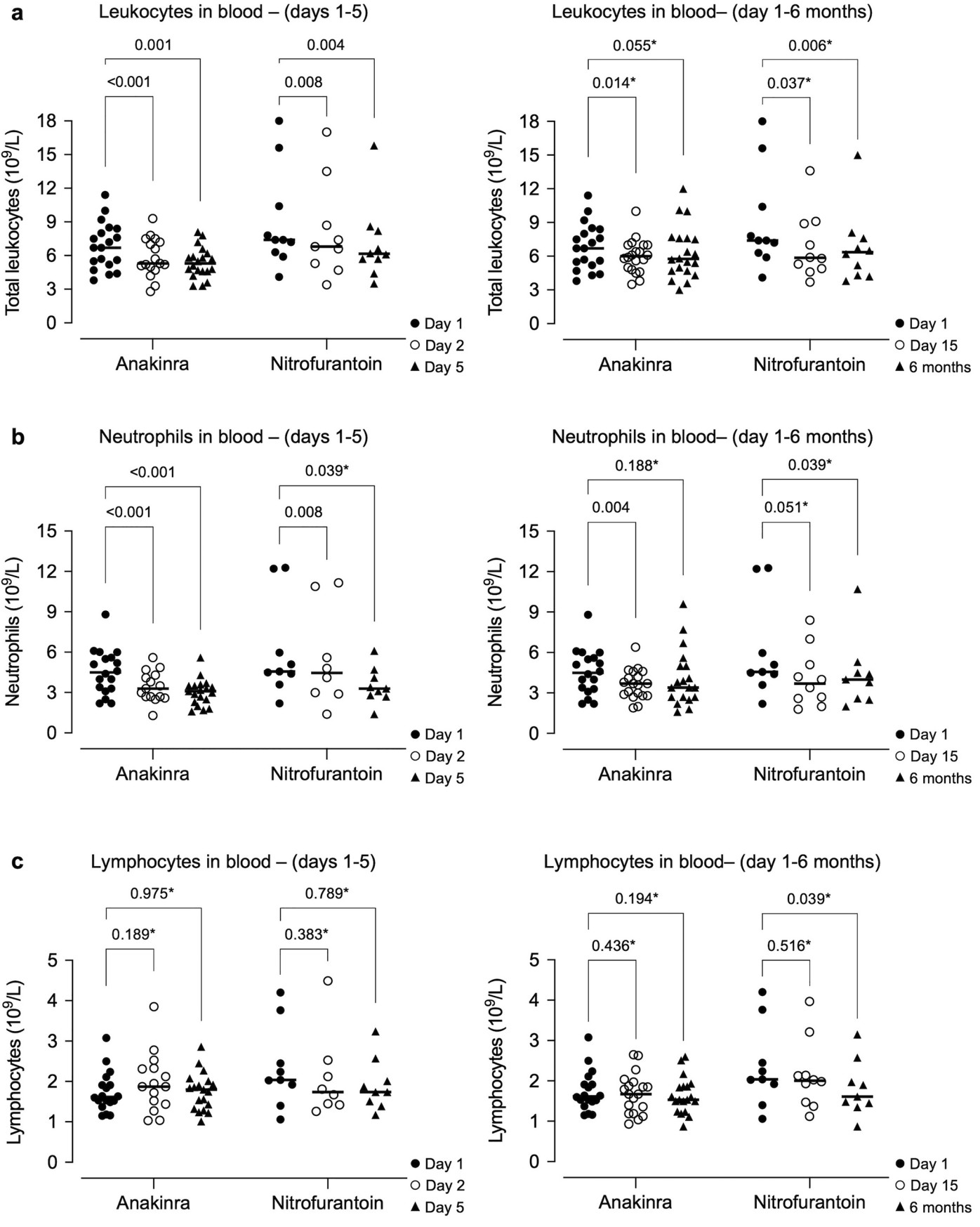

**Extended Data Fig. 1 | See next page for caption.**

**Extended Data Fig. 1 | Total leukocyte and differential neutrophil and lymphocyte counts.** Comparative analysis of blood leukocytes (**a**), neutrophils (**b**) and lymphocytes (**c**) counts in individual patients at enrolment and at different times of follow up. Total leukocyte counts for patients treated with anakinra (n = 20 for Day 5, 15 and 6 months, n = 19 for Day 1, n = 17 for Day 2) or nitrofurantoin (n = 10 for Day 1, 5, 15 and 6 months, n = 9 for Day 2) are shown, as well as differential neutrophil and lymphocyte counts for patients treated with anakinra (n = 19 for Day 5, 15 and 6 months, n = 18 for Day 1, n = 15 for Day 2) or nitrofurantoin (n = 10 for Day 15, n = 9 for Day 1, 5 and 6 months, n = 8 for Day 2). Two-sided $P$-values from post-hoc Wilcoxon matched-pairs signed rank test with two-stage step-up method of Benjamini, Krieger and Yekutieli FDR (Q) = 1% are reported (see also Supplementary Fig. 2 for Aligned Rank Transform (ART) ANOVA test results). *indicates P values that did not pass the FDR threshold. Horizontal lines indicate group median values.

**Extended Data Table 1 | Summary of microbiological data**

| Bacterial species, number of isolates | Day 1 | | Day 5 | | Day 15 | | 6 months | | Total |
|---|---|---|---|---|---|---|---|---|---|
| | Anakinra | Nitro. | Anakinra | Nitro. | Anakinra | Nitro. | Anakinra | Nitro. | |
| *Escherichia coli* | 11 | 5 | 11 | 2 | 9 | 2 | 6 | 4 | 64 |
| Mixed flora | 2 | 3 | 3 | | 3 | 2 | 5 | 1 | 24 |
| *Enterococcus* species | 3 | 3 | 3 | 1 | 2 | | 2 | 2 | 21 |
| *Staphylococcus* species | 4 | 3 | 3 | 1 | 2 | 1 | 1 | | 19 |
| *Streptococcus* species | 2 | | 4 | | 4 | 1 | 3 | 1 | 18 |
| *Klebsiella pneumoniae* | 2 | 1 | 3 | 1 | 2 | | | | 12 |
| *Stenotrophomonas maltophilia* | | 1 | | 1 | | | | | 3 |
| *Citrobacter koseri* | | 1 | | | | | | 1 | 2 |
| *Corynebacterium* species | | | | | 1 | 1 | | 1 | 3 |
| *Lactobacillus* species | | | | | | 1 | | 1 | 2 |
| Other species | | | | 1 | | | 2 | | 3 |
| Positive samples, N(%) | 17(85) | 9(90) | 16(84) | 3(30) | 14(70) | 6(67) | 11(55) | 7(70) | 45 |
| Negative samples, N(%) | 3(15) | 1(10) | 3(16) | 7(70) | 6(30) | 3(33) | 9(45) | 3(30) | 43 |

Summary Table of urine culture data in each treatment group and at each time point. Numbers and per cents of positive cultures are indicated for each species and totally, per group.

**Extended Data Table 2 | Microbiology in the anakinra group**

| Patient | Day 1 Visit 1 | Day 5 Visit 5 | Day 15 Visit 6 | 6 months Visit 7 |
|---|---|---|---|---|
| **P1** | *Escherichia coli* $10^7$ <br> Gram+ mixed flora $10^4$ | *Escherichia coli* $10^7$ <br> *Klebsiella pneumoniae* $10^5$ <br> *Streptococcus anginosus* $10^4$ | *Escherichia coli* $10^6$ <br> *Corynebacterium aurimucosum* $10^5$ <br> Gram+ mixed flora $10^4$ | Gram+ mixed flora $10^4$ |
| **P2** | *Escherichia coli* $10^6$ <br> *Streptococcus agalactiae* (B-strep) $10^4$ <br> *Staphylococcus epidermidis* $10^5$ | *Escherichia coli* $10^5$ | *Escherichia coli* $10^6$ <br> *Streptococcus agalactiae* (B-strep) $10^3$ | *Escherichia coli* $10^7$ <br> *Streptococcus agalactiae* (B-strep) $10^3$ <br> *Enterococcus faecalis* $10^3$ |
| **P5** | *Staphyloccus haemolyticus* $10^4$ | *Staphyloccus haemolyticus* $10^3$ <br> *Escherichia coli* $10^3$ <br> *Streptococcus agalactiae* (B-strep) $10^3$ <br> *Enterococcus faecalis* $10^3$ | *Streptococcus agalactiae* (B-strep) $10^4$ | Gram+ mixed flora $10^3$ <br> *Escherichia coli* $10^3$ |
| **P6** | *Escherichia coli* $10^3$ <br> *Enterococcus* species $10^3$ | *Escherichia coli* $10^5$ | *Escherichia coli* $10^7$ | *Escherichia coli* $10^7$ <br> Gram+ mixed flora $10^3$ |
| **P7** | *Enterococcus faecalis* $10^5$ <br> *Klebsiella pneumoniae* $10^3$ <br> *Streptococcus agalactiae* (B-strep) $10^4$ | *Enterococcus faecalis* $10^4$ <br> *Klebsiella pneumoniae* $10^3$ | *Enterococcus faecalis* $10^5$ <br> *Klebsiella pneumoniae* $10^5$ <br> *Staphylococcus epidermis* $10^5$ | $<10^3$ |
| **P9** | $<10^3$ | $<10^3$ | Gram+ mixed flora $<10^3$ | $<10^3$ |
| **P10** | *Escherichia coli* $10^7$ <br> *Enterococcus faecalis* $10^4$ | *Escherichia coli* $10^7$ <br> Gram+ mixed flora $10^3$ | *Escherichia coli* $10^7$ <br> Gram+ mixed flora $10^3$ | Mixed flora $10^3$ |
| **P11** | *Escherichia coli* $10^4$ | $<10^3$ | $<10^3$ | $<10^3$ |
| **P14** | *Escherichia coli* $10^5$ | *Escherichia coli* $10^5$ | *Escherichia coli* $10^5$ | *Escherichia coli* $10^4$ |
| **P15** | $<10^3$ | Gram+ mixed flora $10^3$ | $<10^3$ | $<10^3$ |
| **P16** | *Escherichia coli* $10^4$ | *Escherichia coli* $10^7$ <br> *Streptococcus agalactiae* (B-strep) $10^3$ | *Escherichia coli* $10^3$ | Mixed flora $10^3$ |
| **P17** | *Staphylococcus saprophyticus* $10^6$ | *Staphylococcus saprophyticus* $10^7$ | *Streptococcus agalactiae* (B-strep) $10^3$ <br> *Escherichia coli* $10^3$ | *Streptococcus agalactiae* (B-strep) $10^4$ <br> *Staphylococcus saprophyticus* $10^4$ |
| **P19** | *Escherichia coli* $10^7$ | *Escherichia coli* $10^6$ <br> Gram+ mixed flora $10^3$ | *Escherichia coli* $10^6$ <br> Gram+ mixed flora $10^3$ | *Escherichia coli* $10^7$ <br> *Actinotignum schaalii* $10^5$ |
| **P21** | *Klebsiella pneumoniae* $10^6$ | *Klebsiella pneumoniae* $10^6$ | *Klebsiella pneumoniae* $10^6$ | $<10^3$ |
| **P23** | *Escherichia coli* $10^7$ | *Escherichia coli* $10^7$ | *Escherichia coli* $10^6$ | *Escherichia coli* $10^7$ |
| **P24** | *Escherichia coli* $10^5$ | *Escherichia coli* $10^6$ | $<10^3$ | $<10^3$ |
| **P26** | *Escherichia coli* $10^7$ | $<10^3$ | Gram+ mixed flora $<10^3$ | $<10^3$ |
| **P27** | *Staphylococcus saprophyticus* $10^5$ | *Staphylococcus saprophyticus* $10^5$ <br> *Streptococcus agalactiae* (B-strep) $10^3$ | *Staphylococcus saprophyticus* $10^4$ <br> *Streptococcus agalactiae* (B-strep) $10^3$ | *Enterococcus faecalis* $10^3$ <br> *Streptococcus agalactiae* (B-strep) $10^3$ |
| **P28** | $<10^3$ | *Enterococcus faecalis* $10^6$ <br> *Escherichia coli* $10^7$ | *Enterococcus faecalis* $10^6$ | $<10^3$ |
| **P30** | Mixed flora $10^3$ | ND | $<10^3$ | $<10^3$ |

Analysis of bacterial numbers and species from individual urine cultures in the anakinra treatment group, at enrolment, after 5 days or 15 days of treatment and at the 6 months follow up visit. The color code is as seen in Extended Data Table 1. Cultures with $<10^3$ CFU/mL were defined as negative. ND = no data.

**Extended Data Table 3 | Microbiology in the nitrofurantoin group**

| Patient | Day 1 Visit 1 | Day 5 Visit 5 | Day 15 Visit 6 | 6 months Visit 7 |
|---|---|---|---|---|
| P3 | *Escherichia coli* $10^6$<br>*Enterococcus faecalis* $10^5$<br>*Citrobacter koseri* $10^4$<br>*Klebsiella pneumoniae* $10^5$ | $<10^3$ | Mixed flora $<10^3$ | *Escherichia coli* $10^5$<br>*Enterococcus faecalis* $10^5$<br>*Citrobacter koseri* $10^5$ |
| P4 | *Escherichia coli* $10^7$<br>Gram+ mixed flora $10^3$ | $<10^3$ | $<10^3$ | *Streptococcus agalactiae* (B-strep) $10^5$ |
| P8 | *Staphylococcus saprophyticus* $10^5$ | *Staphylococcus haemolyticus* $10^4$<br>*Escherichia coli* $10^4$ | *Staphylococcus* (coagulase neg.) $10^3$ | $<10^3$ |
| P12 | Gram+ mixed flora $10^3$ | $<10^3$ | *Streptococcus agalactiae* (B-strep) $10^3$ | *Lactobacillus crispatus* $10^5$ |
| P13 | *Escherichia coli* $10^6$<br>Mixed flora $10^3$ | $<10^3$ | $<10^3$ | $<10^3$ |
| P18 | *Escherichia coli* $10^6$<br>*Enterococcus faecalis* $10^6$<br>*Stenotrophomonas maltophilia* $10^5$ | *Escherichia coli* $10^4$<br>*Enterococcus faecalis* $10^6$<br>*Stenotrophomonas maltophilia* $10^5$ | *Escherichia coli* $10^6$<br>Gram+ mixed flora $10^4$ | *Escherichia coli* $10^7$ |
| P20 | *Staphylococcus saprophyticus* $10^6$ | $<10^3$ | ND | *Enterococcus faecalis* $10^7$ |
| P22 | *Escherichia coli* $10^7$ | *Klebsiella pneumoniae* $10^5$<br>*Providencia rettgeri* $10^4$ | *Escherichia coli* $10^3$ | *Escherichia coli* $10^3$<br>Gram+ mixed flora $10^4$<br>*Corynebacterium striatum* $10^5$ |
| P25 | $<10^3$ | $<10^3$ | *Lactobacillus fermentum* $10^5$<br>*Corynebacterium coyleae* $10^4$ | *Escherichia coli* $10^3$ |
| P29 | *Staphylococcus saprophyticus* $10^5$<br>*Enterococcus faecalis* $<10^3$ | $<10^3$ | Gram+ mixed flora $10^3$ | $<10^3$ |

Analysis of bacterial numbers and species from individual urine cultures in the nitrofurantoin treatment group, at enrolment, after 5 days or 15 days of treatment and at the 6 months follow up visit. The color code is as seen in Extended Data Table 1. Cultures with $<10^3$ CFU/mL were defined as negative. ND = no data.

**Extended Data Table 4 | Adverse events, individual data**

| Patient | Treatment | Type | Onset day | Duration (days) | Severity |
|---------|-----------|------|-----------|-----------------|----------|
| 2 | anakinra | Headache | 3 | 1 | Mild |
| 24 | anakinra | Headache | 5 | 1 | Mild |
| 10 | anakinra | Headache and nausea | 5 | 1 | Mild |
| 7 | anakinra | Nausea and chills | 1 | 3 | Mild |
| 8 | nitrofurantoin | Vomiting | 2 | 1 | Mild |
| 19 | anakinra | Cold with cough | 3 | 1 | Mild |
| 10 | anakinra | Elevated liver enzymes | 1 | 181 | Mild |
| 9 | anakinra | Vaginal rash (HSV-1 positive) | 5 | 27 | Mild |
| 16 | anakinra | Rash at injection site | 13 | 8 | Mild |
| 19 | anakinra | Urticaria | 16 | 19 | Mild |

# Reporting Summary

## Statistics

For all statistical analyses, confirm that the following items are present in the figure legend, table legend, main text, or Methods section.

| n/a | Confirmed | |
|---|---|---|
| ☐ | ☒ | The exact sample size (*n*) for each experimental group/condition, given as a discrete number and unit of measurement |
| ☐ | ☒ | A statement on whether measurements were taken from distinct samples or whether the same sample was measured repeatedly |
| ☐ | ☒ | The statistical test(s) used AND whether they are one- or two-sided *Only common tests should be described solely by name; describe more complex techniques in the Methods section.* |
| ☒ | ☐ | A description of all covariates tested |
| ☐ | ☒ | A description of any assumptions or corrections, such as tests of normality and adjustment for multiple comparisons |
| ☐ | ☒ | A full description of the statistical parameters including central tendency (e.g. means) or other basic estimates (e.g. regression coefficient) AND variation (e.g. standard deviation) or associated estimates of uncertainty (e.g. confidence intervals) |
| ☐ | ☒ | For null hypothesis testing, the test statistic (e.g. *F*, *t*, *r*) with confidence intervals, effect sizes, degrees of freedom and *P* value noted *Give P values as exact values whenever suitable.* |
| ☒ | ☐ | For Bayesian analysis, information on the choice of priors and Markov chain Monte Carlo settings |
| ☒ | ☐ | For hierarchical and complex designs, identification of the appropriate level for tests and full reporting of outcomes |
| ☒ | ☐ | Estimates of effect sizes (e.g. Cohen's *d*, Pearson's *r*), indicating how they were calculated |

*Our web collection on statistics for biologists contains articles on many of the points above.*

## Software and code

Policy information about availability of computer code

| Data collection | Gene expression microarray - GeneTitan System (ThermoFisher Scientific) |
|---|---|
| Data analysis | Statistics - Code used for statistical analysis is available at CodeOcean (doi.org/10.24433/CO.8500789.v1); Prism version 10.5.0 for macOS (GraphPad Software); R version 4.5.1; R Studio version 2025.05.01 Build 513 (Posit Software, PBC); Microsoft Excel for Mac version 16.96 Transcriptomic data - Transcriptome Analysis console (TAC) version 4.0.1.36. (Applied Biosystems); Ingenuity Pathway Analysis Version 01-23-01 (Qiagen) |

For manuscripts utilizing custom algorithms or software that are central to the research but not yet described in published literature, software must be made available to editors and reviewers. We strongly encourage code deposition in a community repository (e.g. GitHub). See the Nature Portfolio guidelines for submitting code & software for further information.

## Data

Policy information about availability of data

All manuscripts must include a data availability statement. This statement should provide the following information, where applicable:
- Accession codes, unique identifiers, or web links for publicly available datasets
- A description of any restrictions on data availability
- For clinical datasets or third party data, please ensure that the statement adheres to our policy

Gene expression data generated in this study are available at NCBI Gene Expression Omnibus (GEO) repository GSE315861. All clinical data supporting the findings

of this study and the study protocol are available in the Article and Supplementary Information. Source data are provided with this paper. All other data that support the findings of this study are available from the corresponding author upon reasonable request. De-identified individual and/or study-level data will be shared with researchers who provide a methodologically sound proposal and if regulatory criteria are met. Access to anonymized data may be granted following review (time frame <20 office days) to ensure compliance with relevant ethical and legal considerations.

# Research involving human participants, their data, or biological material

Policy information about studies with human participants or human data. See also policy information about sex, gender (identity/presentation), and sexual orientation and race, ethnicity and racism.

| | |
|---|---|
| Reporting on sex and gender | Rationale for female patients<br>The disease uncomplicated (recurrent) cystitis occurs only in women. Therefore, only female patients were included in this clinical trial. |
| Reporting on race, ethnicity, or other socially relevant groupings | Ethnicity information was self-reported by the study participants and recorded in the CRF. |
| Population characteristics | Demographic data, medical history, current diagnosis, physical examination and health parameters were recorded by the study physicians at enrollment in the CRF and closely monitored by an external monitor. No significant differences between the two study arms was detected. |
| Recruitment | Participants were recruited at the Clinic for Urology, Paediatric Urology and Andrology in Giessen, Germany, using mailing list of patients from the bladder consultation clinic and advertisement on the study website. All potentially suitable patients were screened. To avoid self biases, potential participants were recruited consecutively from those who presented at screening. Inclusion and exclusion criteria were carefully defined in the study protocol. Written information was presented to the patients, and participation required a signed informed consent. |
| Ethics oversight | Ethical approval was obtained from the German Ethical Review Authority (ethics vote AZ10/21). The trial was conducted in accordance with the principles of the Declaration of Helsinki principles, the International Council for Harmonisation guidelines for good clinical practice, and applicable national laws and regulatory requirements. |

Note that full information on the approval of the study protocol must also be provided in the manuscript.

# Field-specific reporting

Please select the one below that is the best fit for your research. If you are not sure, read the appropriate sections before making your selection.

☒ Life sciences ☐ Behavioural & social sciences ☐ Ecological, evolutionary & environmental sciences

For a reference copy of the document with all sections, see nature.com/documents/nr-reporting-summary-flat.pdf

# Life sciences study design

All studies must disclose on these points even when the disclosure is negative.

| | |
|---|---|
| Sample size | The primary objective of the trial is to evaluate the safety and efficacy of Anakinra treatment for episodes of acute cystitis in patients with recurrent disease. No formal sample size calculation evaluating the power of the trial has been performed. However, a consideration regarding the sample size was made based on previous studies of Anakinra in a murine acute cystitis model.<br>For efficacy, the sample size was based on analysis of symptom scores, inflammatory parameters and bacterial cultures. A sample size of 20 patients in the Anakinra treatment group and ten in the group receiving antibiotics was deemed suitable to achieve criterion for significance (alpha) 0.05 and power 90% using the paired samples 1-tailed t-test. The null hypothesis is H0: mean change in symptom score = 0 and the alternative hypothesis is HA: mean change in symptom score > 0. |
| Data exclusions | No data was excluded. |
| Replication | Replication of clinical study was not relevant. |
| Randomization | All enrolled subjects were randomized to two study treatment groups at Visit 1, the chance for allocation to the anakinra group or nitrofurantoin group was 2:1. Randomization was performed by the central office of the Center for Clinical Trials of the Philipps-University Marburg. |
| Blinding | The administrated Investigational product was then documented together with date, time and study code in the source documentation and eCRF. The analysis of the study parameters was blinded. |

# Reporting for specific materials, systems and methods

We require information from authors about some types of materials, experimental systems and methods used in many studies. Here, indicate whether each material, system or method listed is relevant to your study. If you are not sure if a list item applies to your research, read the appropriate section before selecting a response.

## Materials & experimental systems

| n/a | Involved in the study |
|-----|----------------------|
| ☒ | Antibodies |
| ☒ | Eukaryotic cell lines |
| ☒ | Palaeontology and archaeology |
| ☒ | Animals and other organisms |
| ☐ | Clinical data |
| ☒ | Dual use research of concern |
| ☒ | Plants |

## Methods

| n/a | Involved in the study |
|-----|----------------------|
| ☒ | ChIP-seq |
| ☒ | Flow cytometry |
| ☒ | MRI-based neuroimaging |

## Clinical data

Policy information about clinical studies

All manuscripts should comply with the ICMJE guidelines for publication of clinical research and a completed CONSORT checklist must be included with all submissions.

| | |
|---|---|
| Clinical trial registration | German Clinical Trials Register, DRKS00025964; EudraCT, 2019-004209-28 |
| Study protocol | The trial protocol and statistical analysis plan can be shared upon academic or research request. |
| Data collection | For this randomized, open label, single-center, two-arm, parallel group, Phase II trial, patients were enrolled at the Clinic for Urology, Paediatric Urology and Andrology in Giessen, Germany. The trial was registered on July 27, 2021 at the German Clinical Trials Register #DRKS00025964 (https://drks.de/search/en/trial/DRKS00025964) and the study was conducted between September 2, 2021 and September 3, 2024. Diagnosis and treatment followed an established clinical pathway and mandated routine tests included urinalysis, full blood count and urine and blood sampling for molecular analyses. |
| Outcomes | The primary objective of the trial is to evaluate the safety and efficacy of Anakinra treatment for episodes of acute cystitis in patients with recurrent disease.<br>Primary Endpoint was the reduction in symptom score measured by the Acute Cystitis Symptom Score (ACSS) in the first 5 days.<br>Secondary Endpoints were recurrent UTI episodes until 6 months, bacteriuria and leucocyturia at visits until 26 weeks. |

## Plants

| | |
|---|---|
| Seed stocks | *Report on the source of all seed stocks or other plant material used. If applicable, state the seed stock centre and catalogue number. If plant specimens were collected from the field, describe the collection location, date and sampling procedures.* |
| Novel plant genotypes | *Describe the methods by which all novel plant genotypes were produced. This includes those generated by transgenic approaches, gene editing, chemical/radiation-based mutagenesis and hybridization. For transgenic lines, describe the transformation method, the number of independent lines analyzed and the generation upon which experiments were performed. For gene-edited lines, describe the editor used, the endogenous sequence targeted for editing, the targeting guide RNA sequence (if applicable) and how the editor was applied.* |
| Authentication | *Describe any authentication procedures for each seed stock used or novel genotype generated. Describe any experiments used to assess the effect of a mutation and, where applicable, how potential secondary effects (e.g. second site T-DNA insertions, mosiacism, off-target gene editing) were examined.* |

