## [Peer Review File · Nature Microbiology]

Targeted innate immune inhibition therapy compared to antibiotics for recurrent acute cystitis: a randomized, open-label phase 2 trial

Corresponding Author: Dr Ines Ambite

Version 0:

Reviewer comments:

Reviewer #1

(Remarks to the Author)

The authors addressed Reviewer 2's comments with one exception. The Methods state, "Primary efficacy endpoint was the reduction in ACSS score in the first 5 days," which implies a continuous change score, whereas the analysis uses a cumulative-link (ordinal logistic mixed-effects) model, which treats the outcome as ordinal.

Please revise the primary endpoint description to align with the model: describe it as an ordinal endpoint (specify thresholds and proportional-odds assumption) consistent with the cumulative-link model.

Decision Letter:

Our ref: NMICROBIOL-25104017-T

20th November 2025

Dear Dr. Ambite,

Thank you for submitting your revised manuscript "Non-antibiotic treatment of recurrent acute cystitis in a randomized Phase II trial" (NMICROBIOL-25104017-T). It has now been seen by the original referees and their comments are below. The reviewers find that the paper has improved in revision, and therefore we'll be happy in principle to publish it in Nature Microbiology, pending minor revisions to satisfy the referees' final requests and to comply with our editorial and formatting guidelines.

Unfortunately the original referee #2 was unable to look again and so we recruited a new statistical referee to assess your responses.

Thank you again for your interest in Nature Microbiology Please do not hesitate to contact me if you have any questions.

Sincerely,

Reviewer #1 (Remarks to the Author):

The authors addressed Reviewer 2's comments with one exception. The Methods state, "Primary efficacy endpoint was the reduction in ACSS score in the first 5 days," which implies a continuous change score, whereas the analysis uses a cumulative-link (ordinal logistic mixed-effects) model, which treats the outcome as ordinal.

Please revise the primary endpoint description to align with the model: describe it as an ordinal endpoint (specify thresholds and

proportional-odds assumption) consistent with the cumulative-link model.

Version 1:

Decision Letter:

8th January 2026

Dear Dr Ambite,

I am pleased to accept your Article "Targeted innate immune inhibition therapy compared to antibiotics for recurrent acute cystitis: a randomized, open-label phase 2 trial" for publication in Nature Microbiology. Thank you for having chosen to submit your work to us and many congratulations.

Authors may need to take specific actions to achieve compliance with funder and institutional open access mandates. If your research is supported by a funder that requires immediate open access (e.g. according to [a href="https://www.springernature.com/gp/open-science/plan-s-compliance">Plan S principles](https://www.springernature.com/gp/open-science/plan-s-compliance) or the [a href="https://www.springernature.com/gp/open-science/us-federal-agency-compliance">NIH public access policy](https://www.springernature.com/gp/open-science/us-federal-agency-compliance)) then you should select the gold OA route, and we will direct you to the compliant route where possible. Because authors warrant under our subscription licensing terms that they haven't committed to licensing any version of their article under a licence inconsistent with the terms of our agreement – including the applicable embargo period – publication under the subscription model isn't suitable for authors whose funders require no embargo.

An online order form for reprints of your paper is available at [a href="https://www.nature.com/reprints/author-reprints.html">https://www.nature.com/reprints/author-reprints.html](https://www.nature.com/reprints/author-reprints.html). All co-authors, authors' institutions and authors' funding agencies can order reprints using the form appropriate to their geographical region.

With kind regards,

P.S. Click on the following link if you would like to recommend Nature Microbiology to your librarian
<http://www.nature.com/subscriptions/recommend.html#forms>

** Visit the Springer Nature Editorial and Publishing website at http://editorial-jobs.springernature.com?utm_source=ejP_NMicro_email&utm_medium=ejP_NMicro_email&utm_campaign=ejp_NMicro for more information about our career opportunities. If you have any questions please click [here](mailto:editorial.publishing.jobs@springernature.com).

Open Access This Peer Review File is licensed under a Creative Commons Attribution 4.0 International License, which permits use, sharing, adaptation, distribution and reproduction in any medium or format, as long as you give appropriate credit to the original author(s) and the source, provide a link to the Creative Commons license, and indicate if changes were made. In cases where reviewers are anonymous, credit should be given to 'Anonymous Referee' and the source. The images or other third party material in this Peer Review File are included in the article's Creative Commons license, unless indicated otherwise in a credit line to the material. If material is not included in the article's Creative Commons license and your intended use is not permitted by statutory regulation or exceeds the permitted use, you will need to obtain permission directly from the copyright holder.

Response to reviewers' comments
NMICROBIOL-25104017-T

Thank you again for your positive response to our paper and the opportunity to make final edits.

Enclosed please find an extensively revised version of the manuscript and the check list detailing the changes made, as well as all requested files and information.

Response to Reviewer #1:

Please revise the primary endpoint description to align with the model: describe it as an ordinal endpoint (specify thresholds and proportional-odds assumption) consistent with the cumulative-link model.

The primary end point description has been changed.

“The primary efficacy endpoint was the change in symptom score after 5 days of treatment, measured by the ACSS method.”